# Systemic Mastocytosis and Other Entities Involving Mast Cells: A Practical Review and Update

**DOI:** 10.3390/cancers14143474

**Published:** 2022-07-17

**Authors:** Siba El Hussein, Helen T. Chifotides, Joseph D. Khoury, Srdan Verstovsek, Beenu Thakral

**Affiliations:** 1Department of Pathology, University of Rochester Medical Center, Rochester, NY 14607, USA; 2Department of Leukemia, The University of Texas MD Anderson Cancer Center, Houston, TX 77030, USA; htchifotides@mdanderson.org (H.T.C.); sverstov@mdanderson.org (S.V.); 3Department of Hematopathology, The University of Texas MD Anderson Cancer Center, Houston, TX 77030, USA; jkhoury@unmc.edu

**Keywords:** systemic mastocytosis, well-differentiated systemic mastocytosis, myelomastocytic leukemia, *KIT* gene, mast cell activation syndrome

## Abstract

**Simple Summary:**

In this article, we outline recent updates in systemic mastocytosis (SM) classification, including the bone marrow mastocytosis subtype; and discuss the mast cell leukemia subtype (acute or chronic), the rare variant of SM known as well-differentiated SM (morphologic variant), and other diseases involving mast cells (i.e., myelomastocytic leukemia and mast cell activation syndrome) that may be diagnostically challenging. We also provide a concise clinical update of new adjusted risk stratification models and overview new treatments that were approved for advanced SM (midostaurin, avapritinib).

**Abstract:**

Evidence in the recent literature suggests that the presentation spectrum of mast cell neoplasms is broad. In this article, we elaborate on recent data pertaining to minor diagnostic criteria of systemic mastocytosis (SM), including sensitive testing methods for detection of activating mutations in the *KIT* gene or its variants, and adjusted serum tryptase levels in cases with hereditary α-tryptasemia. We also summarize entities that require differential diagnosis, such as the recently reclassified SM subtype named bone marrow mastocytosis, mast cell leukemia (an SM subtype that can be acute or chronic); the rare morphological variant of all SM subtypes known as well-differentiated systemic mastocytosis; the extremely rare myelomastocytic leukemia and its differentiating features from mast cell leukemia; and mast cell activation syndrome. In addition, we provide a concise clinical update of the latest adjusted risk stratification model incorporating genomic data to define prognosis in SM and new treatments that were approved for advanced SM (midostaurin, avapritinib).

## 1. Introduction

Mast cells were initially reported by Paul Ehrlich in 1879, as multifunctional immune cells of myeloid lineage with several physiologic and pathologic functions [1]. In 1977, Kitamura et al. [2], established the hematopoietic origin of adult mast cells. The ubiquitous tissue distribution, heterogeneity, and plasticity of mast cells places them in an advantageous position to not only guard the immune system but also participate in many biological processes, including maintaining homeostasis [3]. Mast cells normally constitute < 1% of marrow elements; however, increased numbers can be detected in the setting of autoimmune and allergic reactions. At present, it is acknowledged that mast cells directly or indirectly regulate innate and adaptive immune responses by communicating with other cells of the immune system [3]; there is mounting evidence that the local microenvironment directly affects mast cell maturation, phenotype, function, and ability to respond to various stimuli by releasing biologically active mediators [4]. Normal mature mast cells demonstrate a very characteristic immunophenotype with high CD117; intermediate CD9, CD33, and CD71; low CD11b and CD38 expression; and negative human leukocyte antigen-DR isotype (HLA-DR) and CD34 expression. The high affinity IgE receptor (FcεR1) is also constitutively expressed on mast cells.

Systemic mastocytosis (SM) is a highly heterogeneous disease characterized by clonal (usually *KIT* mutants) proliferation and accumulation of abnormal mast cells in one or more extracutaneous organs, for example, the bone marrow, gastrointestinal tract, or liver, with a plethora of distinct presentations ranging from high mast cell burden to organ infiltration and failure. The 2017 [5] and 2022 [6] World Health Organization (WHO) classifications of hematologic malignancies subclassified mastocytosis as cutaneous mastocytosis (CM), SM, and mast cell sarcoma (MCS). Clinically, SM can be further divided into bone marrow mastocytosis (BMM), which was reclassified as a discrete SM subtype in the 2022 WHO classification [6]; indolent; smoldering; and advanced (Table 1). Non-advanced SM subtypes are by far the most frequent and include BMM, indolent SM (ISM), and smoldering SM (SSM). Advanced SM comprises aggressive SM, SM with an associated hematologic neoplasm (SM-AHN), and mast cell leukemia (MCL) (Table 1); advanced SM subtypes are associated with organ damage and poor overall survival. In a recent study, Arock et al. proposed a diagnostic algorithm to differentially diagnose SM based on clinical, histopathologic criteria, and molecular markers [7].

In this article, we aim to present practical points and updates, supported by recent literature, to help hematopathologists, hematologists, and oncologists navigate diagnostically challenging clinical scenarios. We elaborate on the diagnostic criteria of SM reported in the most recent 2022 World Health Organization (WHO) classification [6] (Table 2), including sensitive testing methods for detection of activating mutations in the *KIT* gene or its variants and the diagnostic importance of serum tryptase levels. We summarize the recently reclassified SM subtype BMM, the SM subtype MCL (acute or chronic); the rare morphological variant of SM known as well-differentiated SM (WDSM); myelomastocytic leukemia (MML) that is distinct from MCL; and mast cell activation syndrome (MCAS), which is defined as systemic (involving more than one organ system), severe, recurring mast cell activation. In addition, we present a concise clinical update of the latest adjusted risk stratification model incorporating genomic data to define prognosis in SM, and new treatment options currently available for advanced SM. 

## 2. Updates in Diagnosis and Subclassification of Systemic Mastocytosis

### 2.1. KIT Gene, Hot Spot Mutation (KIT D816V), and Variant Mutations: Current Standards and Suggested Approach for Testing

The *KIT* gene is located on chromosome 4q12, comprises 21 exons, and is composed of four domains (extracellular, transmembrane, juxtamembrane, and an intracellular tyrosine kinase domain that contains a hydrophilic insert of approximately 80 amino acid residues); the extracellular domain consists of five immunoglobulin-like subunits, and the protein kinase domain is interrupted by a hydrophilic insert sequence of about 80 amino acids [9,10,11]. In wild-type KIT, the ligand stem cell factor (SCF) attaches to KIT via the second and third immunoglobulin domains [12], thereby triggering KIT autophosphorylation and dimerization and downstream activation of several pathways; however, *KIT* D816V mutations induce ligand-independent activation of KIT and aberrant mast cell proliferation [13]. The ligand SCF plays a critical role in the regulation of mast cell proliferation and survival. The hotspot mutation in codon 816 (*KIT* D816V) is located in exon 17. *KIT* D816V is detected in more than 90% of SM patients (regardless of subtype) and has a primordial importance because: (1) it is an oncogenic, hallmark “driver” mutation of SM and is considered a minor diagnostic SM criterion according to the 2017 [5] and 2022 [6] WHO classifications; (2) it is linked to aberrant expression of antigens CD25 and/or CD2 [14]; (3) the *KIT* D816V variant allele burden may have significant prognostic value [15,16] and is correlated with the SM subtype; and (4) harboring *KIT* mutations constitutes a decisive factor regarding diagnosis and targeted drug therapy (tyrosine kinase inhibitors, TKIs) [11,17]. Furthermore, the clinical course of mastocytosis is affected by the presence of *KIT* mutations in non-mast cell lineages [7]; the more mast cell lineage-restricted progenitors are affected, the more indolent the disease. Conversely, the more undifferentiated progenitor cells and hematopoietic lineages are affected, the more aggressive the disease [18]: The frequency of *KIT* D816V-mutated non-mast cell lineages (generally myeloid, but occasionally lymphoid lineages) appears to be greater in aggressive SM or MCL as compared to ISM [18]. In one study, multilineage *KIT* D816V involvement was the most important prognostic criterion for progression of ISM to the more aggressive SM subtypes [19]. Besides *KIT* D816V, variant mutations at codon 816 of *KIT* within the tyrosine kinase activation loop, including D816F, D816Y, D816G, and D816I, and mutations at nearby codons, including I817V, N819Y, L799F, D820G, N822L, N822I, InsVI815-816, E839K, S840N, and S849I, have also been reported in SM [6,11,20,21]. In the recent, updated consensus proposal [8] and the 2022 WHO classification [6], besides the *KIT*-activating point mutation(s) at codon 816, mutations in these other critical regions of the *KIT* gene (detected in either the bone marrow or other extracutaneous organs) were added to the minor criteria (Table 2). In addition, *KIT* D816V mutation with a variant allele frequency (VAF) ≥ 10% in the peripheral blood or bone marrow leukocytes is useful as it may indicate high mast cell disease burden, with possible multilineage involvement (B-findings) [8]. Of note, the mutational profile in children who have CM more often than adults is distinct and frequently involves exons 8 and 9 of the extracellular KIT domain [20].

*Testing methodology:* Although currently it is not standard clinical practice to screen for *KIT* mutations other than those involving D816V, with the advent of next generation sequencing (NGS), mutations in *KIT* are investigated using a pre-designed NGS myeloid panel. NGS covers the entire coding region of *KIT*; however, variants at 0.1 to 1% VAF are at the sensitivity limits, mainly due to sequencing-related background error [21]. Given that there is a high correlation between detection of *KIT* mutations (besides the proportion of neoplastic cells in the specimen) and SM diagnosis, enhancing the sensitivity of detection methods is extremely valuable to avoid false-negative results [22]. Sensitivity enhancement can be achieved by enriching for neoplastic mast cells via laser capture microdissection, magnetic bead-based or flow cytometry-based cell sorting [18,23,24], or application of highly sensitive polymerase chain reaction (PCR) techniques, including digital droplet PCR (ddPCR) and allele-specific quantitative PCR (ASO-qPCR), reaching sensitivity as deep as 0.01% VAF [25,26]. Measurement of *KIT* D816V by ddPCR can be used to assess response to TKI treatment and disease progression in *KIT* D816V-mutated ISM or SSM patients [9,27].

SM can be associated with eosinophilia in ~50% of the cases. If no tight clusters of atypical mast cells are identified, no *KIT* mutation or its variants are detected by NGS or other sensitive methods, and only an aberrant mast cell population with expression of CD25, CD2, and/or CD30 is detected, fluorescence in situ hybridization (FISH) testing for *PDGFRA*, *PDGFRB*, and *FGFR1* gene rearrangements is recommended. This testing rules out myeloid/lymphoid neoplasms with eosinophilia as it is not uncommon to find aberrant mast cell populations in these neoplasms by flow cytometry [28].

### 2.2. Serum Tryptase Levels

Tryptase is a serine protease stored and secreted by mast cells during degranulation. A serum tryptase level that is repeatedly elevated (>20 ng/mL) constitutes a minor diagnostic criterion of SM (Table 2) according to the 2017 [5] and 2022 [6] WHO classifications. Serum tryptase levels are elevated in the vast majority of SM patients but can vary widely with notably elevated serum tryptase levels (≥200 ng/mL) seen in aggressive SM and SM-AHN patients in comparison to ISM [29]. Nevertheless, aggressive SM and other advanced subtypes can demonstrate variable, weak tryptase expression from immunostaining; however, true tryptase negativity is rare in untreated SM [28]. In addition, one study revealed that approximately 20% of ISM patients lack mast cell clusters in the bone marrow, and ~30% of ISM patients show a serum tryptase level below 20 ng/mL [30]. Of note, serum tryptase levels are correlated with mast cell activation of all the mast cells in various organs and do not solely reflect neoplastic mast cell activity [31]. Furthermore, normal mast cells demonstrate a range of activation, and the biology of the secretory phenotype and mediator release patterns have not been fully elucidated [32]. 

Diagnosis of SM in the absence of skin involvement is considerably more challenging. It was recently proposed to define BMM separately from ISM when B-findings are absent and the basal tryptase level is below 125 ng/mL [8]; however, if one of the B-findings is detected, and/or the serum tryptase exceeds 125 ng/mL, the patient should be diagnosed with ISM and not BMM [8]. In addition, the prevalence of hereditary α-tryptasemia (HαT), associated with increased basal serum tryptase level and predilection for mast cell activation, has been shown to be significantly higher in SM patients when compared to healthy controls [33]. An adjustment has been proposed in the consensus proposal [8] for patients with HαT; in this case, the basal tryptase level is divided by 1 plus the extra copy numbers of the alpha tryptase gene. For example: if the tryptase level is 30 ng/mL and 1 extra copy of the alpha tryptase gene is found in a patient with HαT, the corrected HαT tryptase level is 15 (30/2 = 15), and thus, it does not constitute a minor SM criterion in this patient. 

Interestingly, correlation between mast cell mediator levels (including serum tryptase among others) and the presence of mast cell mediator-release symptoms as well as SM burden are topics for further investigation. In one study on patients with ISM, mast cell mediator levels correlated considerably with bone marrow neoplastic mast cell burden but not mast cell mediator-release symptoms [34]. In contrast, mastocytosis patients with hereditary α-tryptasemia show considerably higher serum tryptase levels and mast cell mediator-release symptoms, independent of bone marrow mast cell burden [33].

In addition, serum tryptase levels are elevated in a considerable proportion of cases with acute myeloid leukemia (AML), chronic myeloid leukemia, and myelodysplastic syndrome; [35] consequently, in the presence of an associated myeloid neoplasm, this test has limited diagnostic value and thus, it is not considered a minor diagnostic criterion of SM in these scenarios. As observed in one series [36], in cases of WDSM (Section 2.3), the higher levels of cytoplasmic proteases explain both the hypergranulated morphology of neoplastic bone marrow mast cells and the low serum tryptase levels based on the degree of bone marrow and skin involvement by mast cells. These findings also reflect a decreased release of tryptase from mast cells in patients with WDSM, which appears to correlate with the low frequency of *KIT* D816V mutation in these patients [36]. Another factor that can contribute to low serum tryptase levels in patients with WDSM is the decreased number of perivascular bone marrow mast cell aggregates (versus typical SM) [37], delaying tryptase release into systemic circulation.

### 2.3. Well-Differentiated Systemic Mastocytosis (WDSM)

WDSM is a rare and underrecognized variant of SM (~5% of all cases of neoplastic mast cell proliferation) [19,36]. Morphologically, mast cells in this entity are round-shaped and have a round nucleus in the center and a well-granulated cytoplasm, rather than being spindle-shaped, hypogranular, agranular to asymmetrically distributed cytoplasmic granules. 

A limited number of WDSM cases have been reported to date [28,31,36,38,39,40,41,42,43]. Clinically, WDSM has an early onset presentation with familial aggregation, female predominance, and cutaneous involvement. WDSM represents a variant with molecular heterogeneity and a highly variable tumor burden, despite its unique morphologic, immunophenotypic, and clinical features: mast cells are characterized by a distinct well-differentiated morphology, usually lack CD2 and CD25 but show CD30 expression, and low frequency of the typical *KIT* D816V mutation and other exon 17 *KIT* mutations [36]. However, other *KIT* mutations, such as K509I or F522C, can be detected. Features differentiating WDSM from the current defining criteria for SM are displayed in Figure 1.

The two recent WHO classifications (2017 [5], 2022 [6]) do not recognize WDSM as a distinct subtype; rather, WDSM is a morphologic pattern of mast cells that can be observed in any SM subtype (it is rare in BMM, SSM, and MCL) [6,8]. Many cases of WDSM do not meet the current diagnostic criteria of SM because they frequently are *KIT* wild-type or lack exon 17 *KIT* mutations. These WDSM cases are CD25-negative, CD2-negative, or low and usually have low serum tryptase levels (if other SM minor WHO criteria are absent), thus highlighting the challenges in diagnosing SM with well-differentiated morphology [36]. 

On the other hand, aberrant CD30 expression appears to be a recurrent/reliable feature in reported WDSM cases [31]; this raises the question whether CD30 expression should be implemented as evidence of clonality to mitigate CD2 and CD25 negativity in this setting. Alvarez-Twose et al. suggested the following minor diagnostic criteria for WDSM: (1) adult women who started in childhood or with familial aggregation, (2) clustering of bone marrow mast cells in pairs or triplets, (3) expression of CD30 and over-expression of tryptase by flow cytometry analysis, and (4) mutations involving any codon in the *KIT* gene [36]. Furthermore, in one study, CD30 expression was reported to be preferentially expressed in aggressive SM and MCL when compared to ISM [44]. However, in another study, flow cytometry analysis corroborated aberrant CD30 expression on neoplastic mast cells across SM subtypes with no clear association between CD30 expression levels and specific clinicopathological characteristics [45]. In congruence with the previous findings, the 2020 International Working Conference on Mast Cell Disorders, held by experts from Europe and the United States, supported recognition of aberrant CD30 expression on mast cells by either flow cytometry or immunohistochemistry or both as an independent minor criterion of SM besides CD25 and/or CD2 expression (Table 2) [8]. Thus, detection of CD30 on mast cells by either one or both techniques (flow cytometry and immunohistochemistry) has been added to the minor criteria for SM in the 2022 WHO classification [6]. 

Since the addition of the *KIT*-activating point mutation(s) at codon 816 or other critical regions of the *KIT* gene to the minor SM criteria [6], the unique WDSM features can be recognized, especially that a substantial percentage of reported WDSM cases (lack exon 17 *KIT* mutations and harbor *KIT* mutations primarily in exons 8–11) show excellent response to treatment with imatinib [28,31,36,38,39,40,41,42]. Given that well-differentiated morphology of mast cells can be detected in all subtypes of SM (rarely observed in BMM, SSM, and MCL) [6], it has been proposed to add WDSM as a morphologic variant to all classical SM subtypes, e.g., SSM-WDSM for smoldering SM with well-differentiated morphology [8,36]. 

### 2.4. Bone Marrow Mastocytosis

According to Valent et al. [8] and the 2022 WHO classification [6], BMM was reclassified as a discrete SM subtype (in the 2017 WHO classification, it was considered as a special type of ISM [5]) with bone marrow involvement, absence of skin lesions, B-finding(s), dense SM infiltrates in an extramedullary organ, basal serum tryptase < 125 ng/mL, no signs/criteria of MCL, and an AHN [34,46,47,48,49]. BMM is associated with osteoporosis and severe anaphylaxis [34,46,47,48,49]. Patients with BMM and low disease burden (defined as absence of B-findings and a tryptase level < 125 ng/mL) have a superior prognosis to patients with typical ISM or SSM [49]; thus, the importance to accurately diagnose this entity. Of note, in the absence of skin lesions, if B-findings are present and/or serum tryptase is over 125 ng/mL, the disease would be considered ISM and not BMM [8]. For these reasons, BMM was recognized as a separate SM subtype (Table 1) in the recent consensus proposal [8] and the 2022 WHO classification [6]. In BMM, the *KIT* D816V mutation appears to be restricted to mast cell compartments, and allele burden and mast cell infiltrate in the BM are low, occasionally with a well-differentiated morphology, which may explain the indolent course of this entity [48]. Nevertheless, BMM is strongly associated with potentially life-threatening anaphylactic reactions; thus, recognizing BMM early is of paramount importance [46]. 

### 2.5. Mast Cell Leukemia: Chronic versus Acute

MCL is defined as a leukemic subtype of SM when mast cells are ≥20% on bone marrow aspirate smears. These mast cells are immature and atypical with round nuclei rather than spindle-shaped as seen in SM. In a considerable number of patients with MCL, the leukemic spread into peripheral blood is less overt or even absent [50]. In this setting, when mast cells comprise < 10% of all circulating blood leukocytes, the disease is termed “aleukemic” MCL [50]. MCL is further subdivided into primary MCL (absence of prior SM) and secondary MCL (progression following a previous established SM). Furthermore, MCL can be subdivided into acute MCL and chronic MCL when C-findings [8] are present and absent, respectively [50]. Patients with chronic MCL may respond to KIT-targeting drugs and have a better prognosis in comparison to acute MCL. Nevertheless, over time, many patients may progress to acute MCL [51].

Chronic MCL shows similar immunophenotypic patterns to classic cases of MCL, including expression of CD2 (subset), CD25, CD52, CD30, CD117, and tryptase. However, it is good to keep in mind that because some cases of chronic MCL seem to develop in patients with WDSM, neoplastic mast cells in such cases may be CD2- and CD25-negative in contrast to the general immunophenotypic pattern in MCL [50]. More than half (50–70%) of the MCL cases harbor *KIT* D816V. The remainder of MCL cases either harbor other *KIT* mutations (not *KIT* D816V) or lack *KIT* mutations [52,53,54,55], similar to the cases of WDSM, including germline mutations [40] and somatic mutations reported in childhood ISM [20,56]. Based on these findings, it appears that chronic MCL may occasionally develop from a more indolent long-lasting pre-phase of SM with onset at childhood or from WDSM.

However, although both chronic MCL and WDSM demonstrate a slow and long-lasting disease course in the absence of WHO C-findings [8], evidence of long-term history of mastocytosis in the skin has been reported in a substantial number of patients with WDSM [36], a finding that has not been reported in chronic MCL, which characteristically does not affect the skin. Nevertheless, cases of spontaneous regression of skin involvement in some patients prior to bone marrow involvement in WDSM have been reported [36,57], suggesting that chronic MCL can potentially have an undiagnosed phase in which the skin is involved. A key differential diagnosis to chronic MCL is MML, which is discussed in the next section. 

### 2.6. Myelomastocytic Leukemia (MML) 

MML is an extremely rare form of mast cell differentiation in an advanced myeloid neoplasm, e.g., acute myeloid leukemia (AML) or myelodysplastic syndrome with excess blasts, without evidence of CM, SM, MCL, or MCS [58,59]. MML should be differentiated from any SM subtype, including MCL. MML is characterized by >10–19% immature mast cells/metachromatic blasts in the bone marrow and >5% myeloblasts in the bone marrow and/or peripheral blood, wild-type *KIT*, and CD25 negativity [28,58,59]. Differentiating features of MML in comparison to MCL are displayed in Figure 2. 

As MML arises from progression of an underlying myeloid neoplasm, it is frequently associated with a complex karyotype and poor outcome [58,59]. Differential diagnosis of MML includes wild-type *KIT* MCL, SM-AML, chronic eosinophilic leukemia (with increased atypical mast cells), and chronic myeloid leukemia (with increased metachromatic basophils). 

In one retrospective study that reviewed 40 patients who had AML with t(8;21)(q22;q22.1);*RUNX1-RUNX1T1*, 12.5% of the patients were diagnosed with SM or MML when bone marrow biopsies were tested for the *KIT* D816V mutation [60]. In light of these results and to avoid missing an underlying mast cell neoplasm, it is recommended to perform tryptase immunohistochemical stains in all cases of AML with t(8;21). Of note, pre-diagnostic cases of MML have been reported in the literature and include data where the mast cell percentage was in the range 5–9% [58,61,62]. Acute clonal increase in immature atypical mast cells (metachromatic cells) has been conceptualized as secondary and does not meet the criteria of SM-AHN, MCL, or MML [58].

### 2.7. Mast Cell Activation Syndrome (MCAS)

Mast cell activation syndrome (MCAS) is a condition occurring primarily in patients with IgE-dependent allergies and/or mastocytosis; however, it may also be related to other conditions [63]. MCAS is characterized by systemic (involving more than one organ), severe, recurring mast cell activation, typically manifested as severe anaphylaxis (life-threatening events) [64]. The three diagnostic criteria that should be fulfilled to diagnose MCAS include: (1) signs of recurrent, systemic (multiorgan) mast cell activation (e.g., recurrent anaphylaxis); (2) increase in serum mast cell-derived markers of MCAS (tryptase, histamine metabolites, prostaglandin 2 metabolites, and heparin); and (3) symptoms that respond to treatment [63]. MCAS is further classified into (Figure 3): (1) primary/monoclonal MCAS, established by the presence of *KIT* mutations or aberrant expression of monoclonal (often CD25) mast cells [63] (in these MCAS cases, clonal mast cell expansion is obvious without meeting the criteria for SM [65]); (2) secondary MCAS is associated with IgE-dependent allergy or other hypersensitivity reaction but neoplastic mast cells or the *KIT* D816V mutation is not detected; and (3) idiopathic MCAS, for which the criteria to diagnose MCAS are fulfilled; and neither *KIT*-mutations are detected nor is the disease associated with an underlying reactive condition that may explain mast cell activation [64,66]. Treatment with KIT-targeting agents may partially or fully eradicate the mast cell lineage in patients with MCAS [67].

Although MCAS is not a morphologic diagnosis, but rather a clinical and serologic one, pathologists play an important role in investigating evidence of clonality (by detecting *KIT* mutations or aberrant CD25 expression). Once the diagnosis of primary or monoclonal MCAS is established, three entities should be considered and further investigated (Figure 3): (1) CM, (2) SM, and (3) monoclonal MCAS (defined by two minor SM criteria). Although patients who have monoclonal MCAS do not meet the criteria for SM, these patients are considered to have “pre-diagnostic” ISM [68]. Pathologists should be aware of this entity as it can involve classification dilemmas in patients presenting with mast cell activation symptoms with slightly elevated tryptase (<20 ng/mL) and aberrant CD25 expression and/or *KIT* mutation, with absence of aggregates of atypical mast cells [69].

### 2.8. Morphologic Variability of Mast Cells

Contrary to normal mast cells, abnormal mast cells display morphologic variability from round to fusiform, elongated cytoplasmic projections, hypogranular cytoplasm or unevenly distributed granules, and monocytoid morphology [68]. Rare morphologies of mast cells have been described, including spindle-shaped (fibrosarcoma-like pattern), plasmacytoid, epithelioid, histiocytoid, fried-egg like, and anaplastic, rendering their recognition a challenge as these cells are often mistaken for hairy cells, dysplastic eosinophils, basophils, large granular lymphocytes, or promonocytes among other cells [28]. Of note, mast cells with anaplastic morphology and positive expression for CD2, CD25 and CD30 should be further evaluated to rule out anaplastic large cell lymphoma [28].

## 3. Updates in Prognosis and Treatments for Systemic Mastocytosis

The Mayo Alliance Prognostic System (MAPS) is WHO class-dependent and was formed by integrating clinical and hybrid clinical-molecular risk models, aiming to accurately prognosticate SM patients [70]. Following MAPS, the development of a WHO-independent risk model for SM (WHO class-independent MAPS) was based on objective criteria: age > 60 years (two points), platelet counts < 100 × 10^9^/L (two points), hemoglobin levels below the sex-adjusted normal range (two points), increased serum alkaline phosphatase (one point), and serum albumin < 3.5 g/dL (one point) [71]. Patients were distributed into five risk groups: low-risk (zero points, median OS not reached), intermediate-1-risk (1–2 points; median OS = 291 months), intermediate-2-risk (three points, median OS = 99 months), high-risk (4–6 points, median OS = 38 months), and very high-risk (7–8 points, median OS = 8 months). The WHO class-independent MAPS model provides a proof of concept of an achievable WHO class-independent risk categorization of SM patients. The WHO class-independent MAPS model requires further assessment and validation of its performance in the setting of new targeted therapies for SM, such as midostaurin and avapritinib.

The therapeutic approach in ISM patients is primordially directed at anaphylaxis and prevention/symptom control/therapy for osteoporosis; however, patients with advanced SM often need MC cytoreductive therapy to improve disease-related organ dysfunction [68]. Patients with advanced SM who were treated with small molecule kinase inhibitors targeting *KIT* D816V, such as midostaurin [72,73] and avapritinib [74,75], demonstrated high response rates. These medications have changed the therapeutic landscape of the disease. Midostaurin is a multi-kinase inhibitor that the FDA approved in 2017 for advanced SM. Midostaurin targets mutant KIT D816V besides wild-type KIT and other kinases. Avapritinib, which received regulatory approval in June 2021 for advanced SM, selectively and potently targets KIT D816V with a tenfold higher potency compared to midostaurin (IC_50_ = 0.27 versus 2.9 nM, respectively [74,75,76,77,78]. Avapritinib elicited high overall response rates (75%) with median response of 2 months and drastically reduced the mast cell burden (≥50%) and *KIT* D816V VAF in the EXPLORER [79] and PATHFINDER [80] clinical trials, which supported avapritinib’s regulatory approval for advanced SM. A recent systematic literature review of the reports on the trials for avapritinib (EXPLORER and PATHFINDER) and midostaurin (D2201 and A2213 trials), after adjusting for differences in important features of patients with advanced SM, demonstrated the superior efficacy of avapritinib versus midostaurin with respect to overall response rates, disease burden, and survival [81]. Bezuclastinib, a highly selective inhibitor of KIT D816V that exhibits minimal penetrance of the blood–brain barrier is currently being studied in phase 2 clinical trials for ISM, SSM, and advanced SM [82].

Treatment of aggressive SM with imatinib mesylate (multi-kinase inhibitor) was approved in 2006. Imatinib’s therapeutic role is limited to SM patients with wild-type *KIT* [83,84] and imatinib-sensitive transmembrane (F522C) [57] and juxtamembrane (V560G) *KIT* mutations [84,85] as seen in cases of WDSM [86,87]; the kinase domain with *KIT* D816V mutation shows intrinsic resistance to imatinib as certain juxtamembrane mutations do (V559I) [88].

The Mutation-Adjusted Risk Score (MARS) model is a validated, five-parameter, WHO-independent complementary prognostic tool in advanced SM [89]. Patients from the German Registry on Disorders of Eosinophils and Mast Cells and other mastocytosis centers in Europe and the US were included in the study. The MARS model for advanced SM integrates four objective parameters to predict overall survival, namely age > 60 years, hemoglobin < 10 g/dL, platelet counts < 100 × 10^9^/L, and presence of 1 or ≥2 high-risk mutations (e.g., *SRSF2*/*ASXL1*/*RUNX1* or S/A/R) to identify three risk groups: low- (median OS not reached), intermediate- (median OS 3.9 years), and high-risk (median OS 1.9 years) patient groups [89]. The model does not depend on the WHO classification and has been validated; and it may provide guidance for treatment selection and response. For example, the multi-kinase/KIT inhibitor midostaurin demonstrated sustained responses and more favorable outcomes in patients who did not harbor the high-risk mutations *SRSF2*/*ASXL1*/*RUNX1* [72,73,90,91]. Given that most patients are included in the MARS low-risk cohort, midostaurin may be the preferable treatment for these patients [89]. Another registry-based data analysis of patients with advanced SM, based on the MARS parameters and *KIT* D618V VAF, demonstrated the superiority of midostaurin versus cladribine regarding OS and leukemia-free survival [92]. Other studies showed that cytoreduction with cladribine successfully reduced the bone marrow mast cell burden and serum tryptase levels [93,94,95,96,97,98], particularly in patients with *KIT* D816V. A single institution study of 42 patients with advanced SM recently demonstrated that lack of *KIT* D816V mutation was an indication of cladribine resistance [93].

Management of SM-AHN predominantly targets the AHN component, especially if an aggressive disease such as AML is present [77]. Allogeneic stem cell transplant may be suggested in patients with relapsed/refractory advanced SM [68].

## 4. Concluding Remarks

Classification of SM and other diseases involving mast cells has evolved in the last few years given accumulating clinical experience and diagnostic tools to characterize a range of biochemical and genetic features. In this review, we present the SM subtypes BMM and MCL (acute and chronic); the rare morphological variant of SM known as WDSM; the very rare entity named MML that is distinct from MCL; and MCAS. 

Pathologists should be aware of the entities discussed in this review as their diagnoses can be challenging in some cases. WDSM has also been linked to mastocytosis in skin, which suggests that it might represent the evolution of several biologically distinct underlying neoplastic mast cell proliferations. BMM is characterized by bone marrow involvement and no skin lesions and B-findings among other features of SM; for this reason, it was reclassified as a discrete SM subtype (Table 1) in the 2022 WHO classification [6]. Although BMM is associated with an indolent clinical course, it has been strongly linked with potentially life-threatening anaphylactic reactions. MCL cases with >20% mast cells in bone marrow aspirate with a mature morphology and more indolent clinical course are better classified as chronic MCL as they show biologic divergence from the more aggressive typical cases of acute MCL. A key differential diagnosis to chronic MCL is MML. In MML, the criteria for SM are not fulfilled and the neoplastic mast cells (≥10–19% in BM by definition) are thought to be derived from a neoplastic stem cell precursor of an underlying myeloid neoplasm with ≥5% myeloblasts in the bone marrow or peripheral blood [58]. Finally, to avoid the diagnostic pitfall of SM, FISH testing for *PDGFRA*, *PDFGRB*, and *FGFR1* gene rearrangements is recommended to exclude myeloid/lymphoid neoplasms with eosinophilia, as it is not uncommon to find aberrant mast cell populations (with expression of CD2 and/or CD25) in these neoplasms with flow cytometry.

Largely due to sequencing-related background error, sensitivity of *KIT* mutation analysis using NGS is limited to VAFs in the range 0.1–1%. Enriching for neoplastic mast cells or use of highly sensitive ddPCR may enhance the sensitivity of the screening method, reaching VAFs as deep as 0.01% [25,26].

New additions to SM minor diagnostic criteria in the consensus proposal [8] and the 2022 WHO classification [6] include the following (Table 2): (a) activating *KIT* point mutation(s) at codon 816 or in other critical regions of *KIT*; (b) aberrant expression of CD30 by mast cells in the bone marrow, blood, or other extracutaneous organ(s) in addition to CD2 and/or CD25; (c) adjustment of serum tryptase levels > 20 ng/mL in cases of a known HαT.

The WHO class-independent SM prognostic model MARS [89] has been confirmed in an independent validation cohort; this model may provide guidance to select and predict treatment responses in advanced SM. Finally, high response rates were recorded with the small molecule kinase inhibitors midostaurin and avapritinib that target the mutant KIT D816V receptor and wild-type KIT and other kinases (for midostaurin) and were approved for advanced SM. Imatinib mesylate is an efficacious treatment for SM and is limited to patients with wild-type *KIT* and imatinib-sensitive *KIT* mutations in exons 8–11, as seen in WDSM.

## Figures and Tables

**Figure 1 cancers-14-03474-f001:**
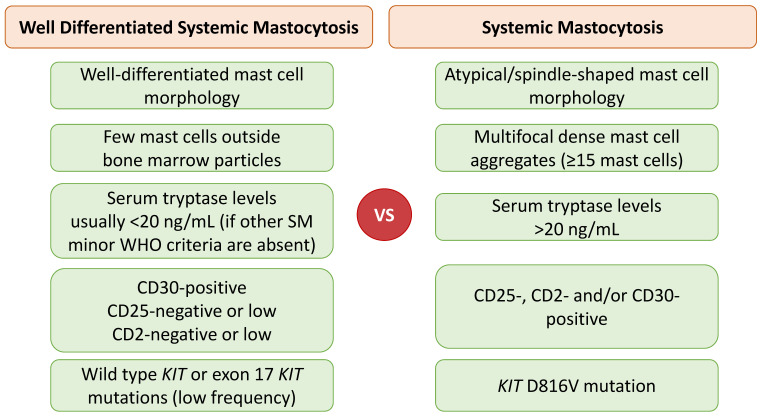
Comparison of current diagnostic criteria in systemic mastocytosis and clinical and pathologic features detected in well-differentiated systemic mastocytosis.

**Figure 2 cancers-14-03474-f002:**
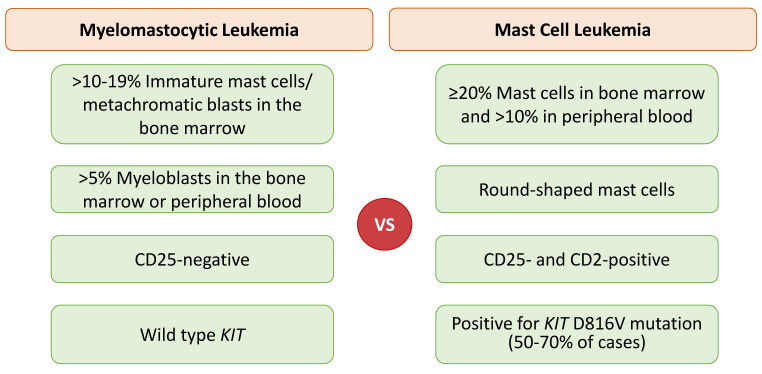
Comparison of diagnostic criteria in myelomastocytic leukemia versus mast cell leukemia.

**Figure 3 cancers-14-03474-f003:**
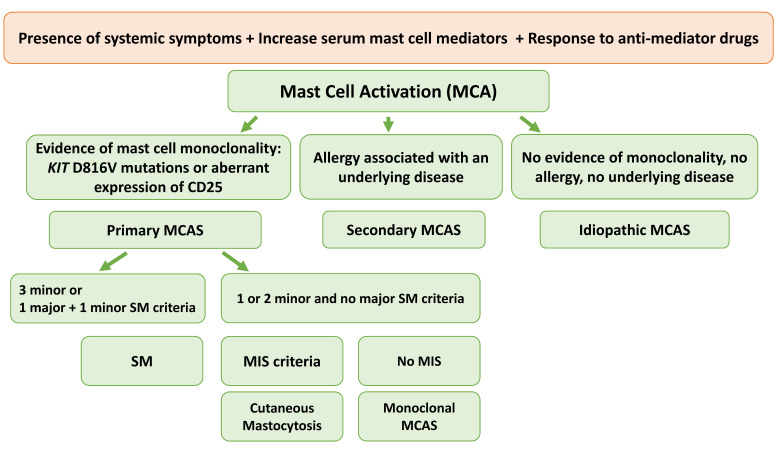
Systematic approach of cases with MCAS (adapted from “Valent P. et al. Proposed diagnostic algorithm for patients with suspected Mast Cell Activation Syndrome. *J Allergy Clin Immunol Pract*. 2019, 7:1125-33 e1.”) [63]. Abbreviations MCAS: mast cell activation syndrome; MIS: mastocytosis in skin; SM: systemic mastocytosis.

**Table 1 cancers-14-03474-t001:** 2022 WHO classification of mastocytosis [6,8].

Cutaneous Mastocytosis (CM) *
**Systemic Mastocytosis (SM) **** Bone Marrow Mastocytosis (BMM) Indolent Systemic Mastocytosis (ISM) Smoldering Systemic Mastocytosis (SSM) Aggressive Systemic Mastocytosis (ASM) SM with an Associated Hematologic Neoplasm (SM-AHN) Mast Cell Leukemia (MCL)
**Mast Cell Sarcoma (MCS)**

***** CM can be further divided into urticaria pigmentosa/maculopapular cutaneous mastocytosis (UP/MCPM) with monomorphic or polymorphic subtypes; diffuse cutaneous mastocytosis (DCM); and cutaneous mastocytoma (isolated or multilocalized). ****** Well-differentiated (WD) morphology may be observed in any SM subtype (rarely in BMM, SSM, and MCL).

**Table 2 cancers-14-03474-t002:** 2022 WHO diagnostic criteria for systemic mastocytosis (SM) [6,8].

For Diagnosis of SM, at Least One Major and One Minor or Three Minor Criteria Are Required.
**Major criterion:** Multifocal dense infiltrates of mast cells (≥15 mast cells in aggregates) detected in sections of the bone marrow and/or other extracutaneous organ(s).
**Minor criteria** a. >25% of all mast cells are atypical cells (type I or type II) on bone marrow smears or are spindle-shaped in dense and diffuse mast cell infiltrates in BM or other extracutaneous organ(s). ^a^ b. Activating *KIT* point mutation(s) at codon 816 or in other critical regions of *KIT* ^b^ in the bone marrow or other extracutaneous organ(s). c. Mast cells in bone marrow, blood, or other extracutaneous organ(s) aberrantly express one or more of the following antigens: CD2, CD25, CD30. ^c^ d. Baseline serum tryptase concentration > 20 ng/mL in the absence of a myeloid AHN. ^d^ In the case of a known HαT, the tryptase level should be adjusted. ^e^

^a^ In tissue sections, an abnormal mast cell morphology counts in both a dense infiltrate and a diffuse mast cell infiltrate. In the bone marrow smear, an atypical morphology of mast cells does not count as an SM criterion when mast cells are located in or adjacent to bone marrow particles. Morphologic criteria of atypical mast cells were referenced in the consensus proposal [8]. ^b^ Any type of *KIT* mutation is considered a minor SM criterion when published solid evidence regarding its transforming behavior is available (an overview of potentially activating *KIT* mutations was provided in the supplementary material of Reference [8]). ^c^ Expression has to be confirmed by either flow cytometry or immunohistochemistry or by both techniques. ^d^ Myeloid neoplasms can lead to increased serum tryptase levels; thus, this criterion does not count in cases of SM-AHN. ^e^ A possible method of adjustment has been proposed for known HαT [8]; the basal tryptase level is divided by 1 plus the extra copy numbers of the alpha tryptase gene. For example, when the tryptase level is 30 and 1 extra copy of the alpha tryptase gene is found, the HαT-corrected tryptase level is 15 (30/2 = 15), and therefore, it is not a minor SM criterion in this case. HαT = hereditary alpha-tryptasemia; SM = systemic mastocytosis.

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
