# Peer review of "Systemic Mastocytosis and Other Entities Involving Mast Cells: A Practical Review and Update"

_cancers, 2022, doi:10.3390/cancers14143474_

Round 1

Reviewer 1 Report

This is a comprehensive review on mastocytosis, which covers very well the current knowledge and literature. There are some points the authors should address to expand on the topic and correct some deficiencies.

Major points:

I understand that the focus of the article is on diagnosis and risk stratification of mast cell neoplasms, but the title of the article suggests that it also covers treatment options. This is also stated in the Abstract, last sentence. Here the article falls short and this deficiency should be corrected.

Authors should for example expand on Bezuclastinib as novel TKI for KIT D816V in phase 2 clinical trials (lines 381-383).

In addition, authors cite the recent paper by Valent et al., Drug-induced mast cell eradication: A novel approach to treat mast cell activation disorders? J. Allergy Clin. Immunol. 149, 1866-1874, 2022 (reference 67). This paper provides a list of KIT D816V targeting tyrosine kinase inhibitors and represents a good starting point for listing treatment options.

The approach by using induced pluripotent stem cells (iPS cells) of mastocytosis patients with KIT D816V mutation (Toledo et al., Blood 137, 2070-2084, 2021; also cited in Valent et al., 2022, see above) represents a further and attractive approach for the identification of novel drugs targeting mast cell neoplasms. The study identified Nintedanib as a potential treatment option (see also Valent et al., 2022).

Minor point: The sentence in line 203-205 is not clear and needs some rewording.

Author Response

Major points:

I understand that the focus of the article is on diagnosis and risk stratification of mast cell

neoplasms, but the title of the article suggests that it also covers treatment options. This is also

stated in the Abstract, last sentence. Here the article falls short and this deficiency should be

corrected.

Authors should for example expand on Bezuclastinib as novel TKI for KIT D816V in phase 2

clinical trials (lines 381-383).

In addition, authors cite the recent paper by Valent et al., Drug-induced mast cell eradication: A

novel approach to treat mast cell activation disorders? J. Allergy Clin. Immunol. 149, 1866-1874,

2022 (reference 67). This paper provides a list of KIT D816V targeting tyrosine kinase inhibitors

and represents a good starting point for listing treatment options.

The approach by using induced pluripotent stem cells (iPS cells) of mastocytosis patients with

KIT D816V mutation (Toledo et al., Blood 137, 2070-2084, 2021; also cited in Valent et al.,

2022, see above) represents a further and attractive approach for the identification of novel drugs

targeting mast cell neoplasms. The study identified Nintedanib as a potential treatment option

(see also Valent et al., 2022).

Authors:

Thank you for this suggestion, we have added on Bezuclastinib  in manuscript and included a recent reference  on it [Siebenhaar, Frank, Jason Gotlib, Michael W. Deininger, Daniel J. DeAngelo, Francis Payumo, George Mensing, Hina Jolin, Jessica Sachs, and Tracy I. George. 2021. 'A 3-Part, Phase 2 Study of Bezuclastinib (CGT9486), an Oral, Selective, and Potent KIT D816V Inhibitor, in Adult Patients with Nonadvanced Systemic Mastocytosis (NonAdvSM)', Blood, 138: 3642].

However, the main focus of this review is on updates on systemic mastocytosis (SM) diagnostic criteria, categories, and prognostic systems. As stated in the last line of the abstract, we discussed only approved medications for SM, we suggest that a detailed discussion of the agents in clinical development are beyond the scope of this review.

Minor point: The sentence in line 203-205 is not clear and needs some rewording.

Authors: Thank you, we have edited this line to make it more clear.

Reviewer 2 Report

El Hussein et al. present an interesting and practical literature review on systemic mastocytosis (SM), in the light of 2022 World Health Organization (WHO) classification of hematologic malignancies. The issue is well introduced and arguments are appropriately presented. The authors perform an update in SM diagnosis, subclassification, prognostication and therapeutic approaches. Manuscript is well written and exhaustively referenced and results suitable for publication.

Author Response

Thank you 

Reviewer 3 Report

The article entitled "Systemic Mastocytosis and other entities involving mast cells: A practical review and update" os a very well-written, easy-to-follow and comprehensive compendium of the current literature and update in the field and provides clinicians with a practical guide on how to differentiate between and diagnose variants of SM and differential diagnoses.

I have no comments regarding the content, just recommend a format and spell check, e.g.

Numbering of literature is not proper in the introduction section.

page 3/line 136: PDGFRB

page 8/line 384: advanced SM

On page 2/line 45 the authors could consider including the high affinity IgE receptor (FceR1) as characteristic immunophenotype for mast cells (although not used in clinical routine) 

Author Response

Comments and Suggestions for Authors

The article entitled “Systemic Mastocytosis and other entities involving mast cells: A practical

review and update”is a very well-written, easy-to-follow and comprehensive compendium of

the current literature and update in the field and provides clinicians with a practical guide on how

to differentiate between and diagnose variants of SM and differential diagnoses.

I have no comments regarding the content, just recommend a format and spell check, e.g.

-Numbering of literature is not proper in the introduction section.

Authors: Thank you, we have edited citations in the introduction

-page 3/line 136: PDGFRB

Authors: We have made necessary edits

-page 8/line 384: advanced SM

Authors: We have made necessary edits

-On page 2/line 45 the authors could consider including the high affinity IgE receptor (FceR1) as

characteristic immunophenotype for mast cells (although not used in clinical routine)

Authors: Thank you, we have added this suggestion